# Comparison of Supervised versus Self-Administered Stretching on Bench Press Maximal Strength and Force Development

**DOI:** 10.3390/sports12040109

**Published:** 2024-04-17

**Authors:** Tim Wohlann, Konstantin Warneke, David G. Behm, Stephan Schiemann

**Affiliations:** 1Institute for Exercise, Sport and Health, Leuphana University, 21335 Lüneburg, Germany; tim.wohlann@uol.de (T.W.);; 2University Sports Centre, Carl of Ossietzky University Oldenburg, 26129 Oldenburg, Germany; 3Institute of Movement Science, Sport and Health, Karl-Franzens University Graz, 8020 Graz, Austria; 4School of Human Kinetics and Recreation, Memorial University of Newfoundland, St. John’s, NL A1C 5S7 P.O. Box 4200, Canada; dbehm@mun.ca

**Keywords:** stretching, supervised, home-based, maximal strength, explosive strength

## Abstract

Purpose: While there is reported superior effectiveness with supervised training, it usually requires specialized exercise facilities and instructors. It is reported in the literature that high-volume stretching improves pectoralis muscles strength under supervised conditions while practical relevance is discussed. Therefore, the study objective was to compare the effects of volume equated, supervised- and self-administered home-based stretching on strength performance. Methods: Sixty-three recreational participants were equally assigned to either a supervised static stretching, home-based stretching, or control group. The effects of 15 min pectoralis stretching, 4 days per week for 8 weeks, were assessed on dynamic and isometric bench press strength and force development. Results: While there was a large magnitude maximal strength increase (*p* < 0.001–0.023, ƞ^2^ = 0.118–0.351), force development remained unaffected. Dynamic maximal strength in both groups demonstrated large magnitude increases compared to the control group (*p* < 0.001–0.001, d = 1.227–0.905). No differences between the intervention group for maximal strength (*p* = 0.518–0.821, d = 0.101–0.322) could be detected. Conclusions: The results could potentially be attributed to stretch-induced tension (mechanical overload) with subsequent anabolic adaptations, and alternative explanatory approaches are discussed. Nevertheless, home-based stretching seems a practical alternative to supervised training with potential meaningful applications in different settings.

## 1. Introduction

In several sports and rehabilitation settings, increasing or restoring strength capacity is of paramount importance [1,2] which is commonly achieved using resistance training [3,4]. Nevertheless, even though highly effective, there are a number of difficulties with common resistance training programs such as travelling to specialized training facilities to receive professional supervision. The lack of training success might not be attributable to the effectiveness of resistance training interventions per se, but to the participants limited motivation and commitment to travel to training locations or perform exhausting interventions [5,6]. The literature points out the high demand for time- and space-saving exercise alternatives which can be integrated into the participants or patients daily routines [7,8]. 

Even though the literature reports alternatives, such as blood flow restriction training [9] or electromyostimulation [10,11] to induce sufficient stimuli to improve strength, these still require expensive equipment or coaches which might be not available to the broad population. Potentially using stretching as an alternative was suggested by Arntz et al. [12] and Panidi et al. [13] who reported that high-volume and/or high-intensity stretch training could potentially induce improvements in strength capacity. Accordingly, six weeks of one-hour daily self-administered calf muscle stretching induced increases in maximal strength, muscle thickness, and flexibility that were not significantly different from a commonly used resistance training routine (5 × 12 repetitions on 3 days per week for 6 weeks) [14]. Nevertheless, the plantar flexors can be considered a comparatively small muscle group with comparably low impact on multi-articular, complex (athletic) movements [15,16]. While Wohlann et al. [17], Ikeda and Ryushi [18] and Chen et al. [19] reported stretch-induced strength increases in the thigh muscles, Reiner et al. [20], Warneke et al. [21] and Wohlann et al. [17] showed transferability to the upper body. Wohlann et al. [17] pointed out that 15 min of supervised stretching has the potential to substitute high-intensity pectoralis resistance training. However, Schoenfeld et al. [22] highlighted the impracticality of stretching-induced strength gains, especially when this type of training requires a second person or special equipment. Therefore, this study explores the possibility of alternative and more practical stretching training such as home-based stretching training and directly compares it to supervised stretching training. It is investigated whether home-based stretching training can achieve an equivalent increase in strength capacity as supervised stretching training.

To account for highly specific testing conditions [23,24], strength was tested under isometric and dynamic conditions, as most studies focused on one of these parameters [21,25,26]. While Arntz et al. [12] were not able to detect significant stretch-induced force development enhancements in their meta-analysis, this result might be attributable to the inclusion of short stretching protocols in their analysis. Assuming a dose–response relationship for maximal strength, it was hypothesized that longer stretching durations could be sufficient to affect force development capacities.

## 2. Materials and Methods

Participants from all groups visited the lab three times, which included an initial briefing and a pre- and a post-test. The briefing visit simultaneously served as a familiarization session to avoid adaptations due to learning effects in order to optimize the exercise execution, especially for participants who did not regularly perform maximal repetitions in the bench press. Furthermore, the familiarization session would improve the validity of the isometric maximal strength testing [24]. During both the pre-test and post-test, measurements were taken in the following sequence: isometric, dynamic maximal strength, and force development.

### 2.1. Participants

The required sample size was estimated via G-Power with an estimated effect size of f = 0.25. A total sample size of 42 was estimated. To account for potential dropouts and enhance statistical power, 63 recreationally active participants were recruited from the university sports center and assigned to supervised stretching with a stretching device (SVS), self-administered home-based stretching (HBS), or a control group (CG) (Table 1). The following eligibility criteria were applied: Participants were considered recreationally active when they were physically active at least twice a week without any injuries or surgery in the chest or shoulder during the last 6 months leading to prolonged immobilization and thus training interruptions. Furthermore, as the training program might be primarily applicable to untrained and sedentary populations, flexibility-trained participants were excluded. All participants provided written informed consent at the habituation session.

### 2.2. Maximal Strength and Force Development Tests

Before conducting the maximal strength and force development tests, a standardized warm-up was performed using 5 min of ergometer cycling with 60 revolutions per minute followed by 2 × 5 push-ups for the males and 2 × 5 push-ups with hands on an elevated surface for the females. Afterwards, participants were allowed to perform their individual bench-press warm-up programs, if needed. The bench press movement was performed using a Smith machine (Train Hard, Hansson Sports, Steinbach, Germany)

For the isometric testing condition, the bar was fixed in the Smith machine to provide an unsurpassable (immovable) resistance. The elbow angle of 90° was ensured via goniometer testing. To measure maximal isometric strength, the participants were instructed to push the barbell with maximal effort against the fixed bar. Applied forces were quantified via a Kistler force platform with four 9051 load cells, operating at a sampling frequency of 1000 Hz and connected with an A/D converter NI6009 (National Instruments DAQ 700). The participants performed at least three trials until strength values decreased. A 120 s rest period between each trial was ensured to avoid fatigue. After isometric testing, the dynamic one-repetition-maximum (1 RM) bench press test was conducted. The barbell was loaded with weight until a valid repetition could no longer be performed. A repetition was considered valid when the elbows were positioned below the upper body during the eccentric phase and pushed upward until the elbows were extended without assistance.

For the force development tests, 50% of the 1 RM was used. The barbell was positioned on metal coil springs integrated into the Smith machine, guaranteeing that participants’ elbows remained fixed at a 90° angle as they kept their hands on the barbell. Responding to an acoustic signal, the participants were instructed to perform a pressing movement with the intention to throw the barbell concentrically upward from the chest as quickly as possible to ensure maximal bar velocity. However, for safety reasons, the participants did not actually throw the bar. Impulse (*p* = F × ∆t) was used to interpret the force development behavior and was calculated as follows: Each individual force value (F) within an interval of ∆t (0.001 s) was multiplied, and the sum of these values over the interval was computed. The intervals from the start of contraction to 200 ms and 500 ms were considered for interpretation. Figure 1 shows a force–time curve with force development determination.

### 2.3. Intervention

All participants in the SVS and HBS groups performed an eight-week stretch training program, four days per week with equalized stretching volumes. The SVS group underwent 15 min of passive static stretch training on a custom-made stretching board [17]. Each SVS stretch training session was performed with an examiner. The elbows were fixed at a 90° angle using an orthosis, while the shoulder angle was maintained at 90° to achieve maximum stretching of the pectoralis major muscle. An automatic ratchet was used to retighten continuously to counteract relaxation effects, which are assumed to decrease resistive force in constant-angle stretching. To prevent any excessive arching of the back, participants positioned their legs against a wall (Figure 2). Participants in the HBS group followed a standardized 3 × 5 min static stretching for the chest muscles with three stretching exercises as a home-based training using a standardized resistance band, identical to that in Warneke et al. [21]. The stretching exercises for the HBS group were carried out independently by the participants, while the stretching duration and adherence were documented in a stretching diary. Stretching intensity was set to the maximum-tolerated stretching pain.

### 2.4. Data Analysis

Statistical analysis was carried out using IBM SPSS Statistics version 28 (IBM SPSS, version 28). A normal distribution of the main outcome data was ensured using the Kolmogorov–Smirnov test (n > 30) and the homogeneity of variance was ensured with the Levene test. The data are presented as mean (M) and standard deviations (SDs). Reliability is expressed via intraclass correlation coefficients (ICCs) and coefficients of variance (CVs). A one-way analysis of variance (ANOVA) was conducted to test for pre-test group differences, while the research question was evaluated via two-way repeated-measures ANOVA (3 groups × 2 testing times) with a Scheffé post hoc analysis. Between-group differences were reported using the following effect size classifications: small effect (d < 0.5), medium effect (0.5–0.8), and large effect (d > 0.8) [27]. The critical significance level was set at *p* = 0.05.

## 3. Results

In accordance with Koo and Li [28], ICCs ranging from 0.96 to 1, CV = 0.2–3.6% for isometric and dynamic maximal bench press strengths, and force development values after 200 ms and 500 ms were classified as high. With *p* > 0.05, a normal distribution was assumed, while the one-way ANOVA ruled out pre-test differences (*p* > 0.05).

### 3.1. Isometric and Dynamic Bench Press

With a time effect of *p* < 0.001 and ⴄ_p_^2^ = 0.23–0.45, both isometric and dynamic testing conditions showed a significant strength increase with a moderate-magnitude Time×Group interaction in the isometric (*p* = 0.023, ⴄ_p_^2^ = 0.118) and a large-magnitude Time×group interaction effect in dynamic testing conditions (*p* < 0.001, ⴄ_p_^2^ = 0.351) (Table 2).

Post hoc testing revealed a significantly greater isometric force with SVS versus CG (*p* = 0.032, d = 0.63) but no differences between HBS and CG (*p* = 0.125, d = 0.53). Dynamic maximal strength showed significant increases in the SVS compared to CG (*p* < 0.001, d = 1.23) and in HBS compared to CG (*p* = 0.001, d = 0.91). No significant differences could be detected between the SVS and HBS in isometric (*p* = 0.821, d = 0.101) and dynamic (*p* = 0.518, d = 0.322) testing conditions, respectively.

### 3.2. Force Development

Neither the Time (*p* = 0.117–0.159) nor the Time×Group interaction (*p* = 0.604–0.619) reached the level of significance, showing force development as remaining unaffected by both stretching conditions (Table 3).

## 4. Discussion

The present study compared the effects of high-volume supervised stretching training with a self-administered equal-volume stretch training on strength performance. Both training conditions significantly increased strength with no superior effectiveness between supervised and non-supervised stretch training. Irrespective of the group, the rate of force development determined after 200 ms and 500 ms remained unaffected (*p* = 0.60–0.62).

The study results are in accordance with a growing body of evidence showing high-volume stretch training to sufficiently enhance maximal strength [12,29]. Assuming a dose–response relationship, recent research enhanced the stretching duration up to 2 h per day for 6 weeks [30], showing highly consistent results in the plantar flexors. Similarly, upper-body-muscle static stretching induced pectoralis muscle hypertrophy [17] and strength increases [17,20,21].

### 4.1. Potential Underlying Mechanisms to Explain Stretch-Mediated Strength Increases

Strength increases are commonly explained with morphological and/or neuromuscular adaptations [31]. Although Goldspink and Harridge [32] suggested that the striated muscle cross-sectional area reflects force production potential, previous studies did not obtain a meaningful relationship between stretch-mediated hypertrophy and strength increases induced via stretching [17,33].

Consequently, a neuronal influence should be considered a potential explanation for stretch-induced strength increases. Adaptations in neuromuscular control were suggested more than 10 years ago by Nelson et al. [26], finding a contralateral force transfer to the non-stretched control leg. However, participants seemed to be untrained, as the authors speculated that stabilization via the non-stretched leg while performing 4 × 30 s stretching on 3 days per week might have caused these increases. However, it is also possible that several reflex mechanisms induced by stretching [34] affected central nervous control, which could be reflected by increases in the contralateral strength [35,36]. Nevertheless, since EMG activity while performing 10 min of static stretching was not significantly enhanced [37], the possibility of substantial neural adaptations is called into question. Furthermore, authors speculated that an elongated muscle could induce muscle contractions against the stretch device that could initiate a training stimulus that might be comparable to full ROM resistance training [38]. 

A further explanation is related to blood flow conditions. Since blood flow restriction training seems to enhance strength capacity and muscle mass with lower-intensity contractions [39], similar adaptations might be possible with prolonged static stretching. Interestingly, McCully [40] investigated blood flow patterns when performing 10 min of stretching and showed restricted blood flow to the muscle. However, since the influence of stretch-induced blood flow restriction on muscle hypertrophy was not explored and no neuromuscular adaptations (i.e., EMG testing, blood flow, muscle hypertrophy) were measured, this rationale remains speculative.

### 4.2. Supervised versus Self-Administered

The strength increases of the different stretching training in this study are in line with other studies, showing daily self-administered stretching in the calf muscles [14] and 15 min supervised continuous pectoralis stretching [17] induced similar strength increases. Wohlann et al. [17] showed comparable increases of those expected by resistance training in untrained populations. A potential advantage of supervised stretching training over self-administered stretching training might be the possibility of ensuring proper exercise execution and, thus, training intensity. In the literature, stretching intensity is often controlled using a visual analog scale (VAS) without quantifying the actual tension on the muscle. Quantifying stretching intensity seems even more crucial considering that Lim and Park [41] found no correlation between measured passive tension and a subjective pain scale. Wohlann et al. [17] showed a continuous decrease in mechanical stretching tension in the intervened muscle (due to relaxation effects) when using constant-angle stretching. Thus, to ensure more constant tension and therefore higher intensities, an adjustment of mechanical tension might be beneficial. However, this might not be applicable in a self-administered stretching routine. Nevertheless, no differences were found between the two stretching groups, indicating a higher practical relevance of the self-administered stretching training due to its independence from location and a second person.

### 4.3. Contraction Specificity

Most studies focused on either isometric or dynamic testing routines. Warneke et al. [24] as well as James et al. [42] underlined specific testing conditions in maximal strength testing, as maximal isometric and dynamic strength should be considered individual abilities. Therefore, assuming movement training specificity, static stretching is more related to isometric testing conditions, and thus a higher increase in isometric strength could be speculated. However, Warneke et al. [33] showed isometric strength to increase about 16%, whereas dynamic strength was enhanced by 25%. 

Furthermore, angle specificity in isometric testing should be considered [24]. Accordingly, Yahata et al. [43] showed strength increases exclusively in the neutral joint angle position, while the plantar flexed isometric testing revealed no pre–post change via stretching. It can be speculated whether the stretching could have led to a change in muscle fiber length and thus a change in joint configuration during movement execution. Panidi et al. [13] demonstrated that stretching interventions with high intensities could lead to a change in muscle fiber length (*p* = 0.006, SMD = 0.28), but may not result in a change in the pennation angle.

Assuming isometric maximal strength measurements do not automatically predict dynamic performance due to different activation patterns of motor neurons [44,45] the present study included both isometric and dynamic testing conditions, which was supplemented by the rate of force development values after 200 ms and 500 ms. However, there were no changes in the rate of force development after 8 weeks of stretching.

### 4.4. Practical Applications

This study was performed to counteract methodological limitations described by Schoenfeld et al. [22] and others [5,6,14], indicating that long stretching durations were impractical. While increasing strength may potentially be particularly relevant for sport-specific tasks such as jumping and sprinting [46], or ball throwing velocity in handball [47], a recently published systematic review did not find stretch-induced performance enhancement [48], which seems in accordance with the lack of results for the rate of force development and explosive strength parameters obtained in the current study [15]. Furthermore, in rehabilitation, there is a high relevance of restoring muscle strength after prolonged phases of immobilization [49] or reduced physical activity. Especially in sedentary populations, the recent literature pointed out the possibility of using high-volume stretch training [8] and referred to studies using prolonged stretching training [30]. Resistance training is efficient, but it is location-dependent and requires special equipment, while supervision by a movement expert is highly recommended, especially for training beginners and on a recreational level. Therefore, the relevance for orthopedic patients with limited mobility, as well as for those with restricted time or lack of motivation, should be considered. This study showed self-administered stretching to be a valid alternative for strength increases, as it could be performed while watching TV or working at the computer [8], without meaningful reductions in effectiveness.

However, whether stretching is a long-term alternative to other training routines remains speculative, as no studies could be found exceeding intervention periods of 8 weeks. Since it is well known that especially untrained and recreationally active participants respond to almost all novel stimuli with strength increases, further research is necessary to validate especially home-based stretching programs for the alternative application in sports practice (≥8-week intervention periods).

### 4.5. Limitations

Even though this study provided further evidence for stretch-induced maximal strength increases, no underlying mechanisms were explored in the present study. Strength increases might be explained by neuromuscular activity changes; however, no EMG study measurements were performed. When testing maximal isometric strength, angle specificity was assumed. Nevertheless, this study used just one given elbow angle, which may be of limited validity for other joint angles. Based on the results of Yahata et al. [43], it can be assumed that different joint angle positions may yield different outcomes. Therefore, the transferability of the results to other joint angle positions needs to be examined. Further research is needed to clarify the underlying mechanism and identify moderators such as stretching intensity, training frequency, or joint angle specificity to assess a best practice model.

In the home-based group, no control of the intensity could be carried out. Therefore, a placebo effect cannot be entirely ruled out. Since Apostolopoulos et al. [50] underlined the relevance of stretch intensity, the lack of control might have limited the results. Nevertheless, no significant difference between the interventions was observed.

## 5. Conclusions

A comparison between self-administered stretching training and supervised stretching training with the same stretching volume has not yet been conducted. Both supervised and self-administered stretching increased bench press maximal strength without a difference between the training modes. The supervised stretching required a second person, organizational coordination, and a special setup to stretch the chest muscle. In contrast, the self-administered stretching could be performed independently by participants at home, regardless of location, time of day, or the need for a second person. A self-administered stretching routine thus appears to be a valid alternative to supervised stretch training when aiming to enhance maximal strength. The results of this study contribute to the discussion on the practicality of stretching training and open perspectives for further practical applications.

## Figures and Tables

**Figure 1 sports-12-00109-f001:**
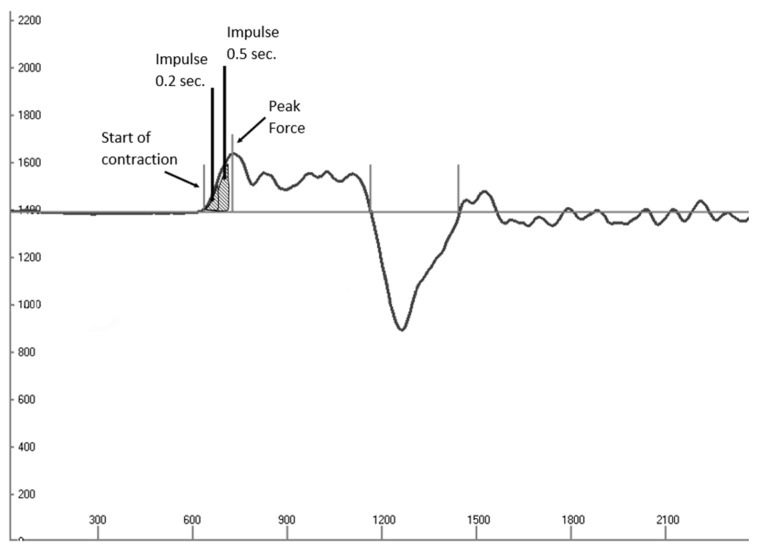
Force–time curve with 50% of 1 RM. Y-axis = measured force in Newtons, x-axis = time in milliseconds. Force development was determined 200 ms (impulse, 0.2 s) and 500 ms (impulse, 0.5 s) after the start of contraction. The straight light gray line represents the calibration and consists of the subject’s body weight, the barbell (115 Newton), and 50% of the weight used in the 1 RM test. The curved dark gray line represents the force output of the participants during the bench press movement.

**Figure 2 sports-12-00109-f002:**
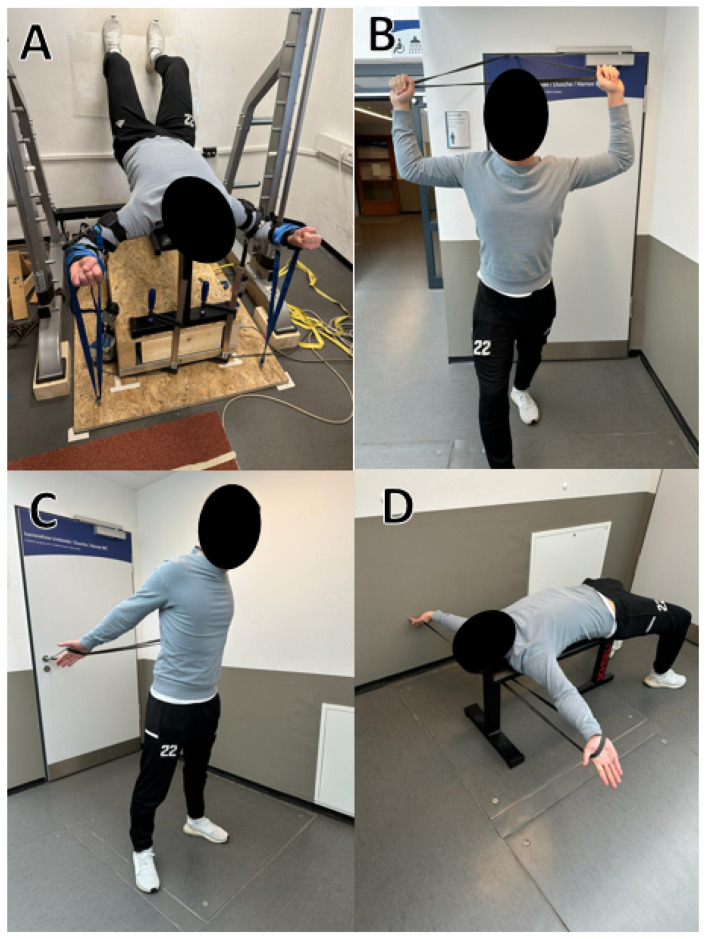
Stretching exercises. (**A**) A period of 15 min of supervised static stretching, (**B**–**D**) home-based stretching exercises, holding for 5 min per session respectively.

**Table 1 sports-12-00109-t001:** Characteristics of the participants.

Group	N (Male/Female)	Age	Height (cm)	Weight (kg)
SVS	21 (13/8)	24.2 ± 2.4	177.9 ± 9.8	73.8 ± 15.2
HBS	21 (13/8)	24.4 ± 3.8	179.6 ± 7.7	76.4 ± 12.2
CG	21 (13/8)	24.3 ± 2.9	177.6 ± 8.8	74.6 ± 11.7

SVS = supervised stretching; HBS = home-based stretching; CG = control group.

**Table 2 sports-12-00109-t002:** Descriptive statistics and two-way ANOVA of isometric and dynamic maximal strength.

Maximal Strength	Group	Pre-Test (m ± SD)	Post-Test (m ± SD)	Change (m ± SD)	Time Effect	Time × Group
Isometric	SVS	559.4 ± 234.1 N	607.1 ± 249.3 N	+8.5%	*p* < 0.001	*p* = 0.023
	HBS	582.2 ± 253.0 N	619.4 ± 267.0 N	+6.4%	F = 18.191	F = 3.997
	CG	571.4 ± 326.1 N	573.9 ± 234.0 N	+0.4%	ⴄ_p_^2^ = 0.233	ⴄ_p_^2^ = 0.118
Dynamic (1 RM)	SVS	61.1 ± 20.6 kg	65.7 ± 22.1 kg	+6.9%	*p* < 0.001	*p* < 0.001
	HBS	62.6 ± 26.1 kg	65.8 ± 26.4 kg	+4.9%	F = 48.666	F = 16.253
	CG	63.6 ± 24.4 kg	63.3 ± 24.2 kg	−0.5%	ⴄ_p_^2^ = 0.448	ⴄ_p_^2^ = 0.351

SVS = supervised stretching; HBS = home-based stretching; CG = control group.

**Table 3 sports-12-00109-t003:** Descriptive statistics and two-way ANOVA of force development.

Force Development	Group	Pre-Test (m ± SD)	Post-Test (m ± SD)	Change (m ± SD)	Time Effect	Time × Group
Impulse 0.2 (N*s)	SVS	235.0 ± 60.2	238.8 ± 60.7	+1.7%	*p* = 0.117	*p* = 0.604
	HBS	246.4 ± 64.7	247.2 ± 62.6	−0.2%	F = 2.526	F = 0.508
	CG	236.6 ± 53.7	238.3 ± 56.4	+0.5%	ⴄ_p_^2^ = 0.040	ⴄ_p_^2^ = 0.017
Impulse 0.5 (N*s)	SVS	564.0 ± 141.8	572.6 ± 144.6	+1.5%	*p* = 0.159	*p* = 0.619
	HBS	596.1 ± 145.3	597.6 ± 144.2	−0.1%	F = 2.033	F = 0.484
	CG	577.1 ± 135.2	580.3 ± 138.6	+0.4%	ⴄ_p_^2^ = 0.033	ⴄ_p_^2^ = 0.016

SVS = supervised stretching; HBS = home-based stretching; CG = control group.

## Data Availability

The datasets used and/or analyzed during the current study are available from the corresponding author on reasonable request.

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
