# Peer review of "Comparison of Supervised versus Self-Administered Stretching on Bench Press Maximal Strength and Force Development"

_sports, 2024, doi:10.3390/sports12040109_

Round 1

Reviewer 1 Report

Comments and Suggestions for Authors

This study examined the effects of supervised and selfadministered stretching of the pectoralis muscles on bench press maximal strength and force development. The research question has some practical applications and adds to the body of literature showing a positive effect of stretching on muscle strength. The text is well well prepared. The introduction contains most of the the relative new references and it provides a nice story ending with a logical hypothesis. Methods are simple but informative with appropriate statistics. The results are clearly presented. No additional information is needed. The discussion focused both on the study limitations, the possible mechanisms behind the phenomenon and the practical applications. Conclusions are based on the current data.  

Author Response

Dear Reviewer,

thank you for your work! 

Best regards,

The Authors

Reviewer 2 Report

Comments and Suggestions for Authors

This is a well-written and well-presented study. Although I consider myself quite experienced, I must admit that I wasn’t aware that simply passive (al be it with a high dose) stretching could lead to force increases (probably only in otherwise untrained individuals?, I hope to read in the discussion) The authors convincingly show that it is possible to get similar results with supervised versus unsupervised training.

-Please indicate what the adherence was in supervised and unsupervised conditions. This is important since adherence always is a problem.

-How were the participants instructed to determine (increase) stretch intensity over time (both within a session as well as across the 8 weeks). In the discussion, I read that such an increase in intensity did not occur in the at-home situation…..this seems remarkable, I missed the mentioning of a placebo effect. I think this should be mentioned, although the authors probably are convinced that the effect is ‘real’, placebo effects also are real and cannot be ruled out.

I read the potential mechanisms with interest and attention: Indeed, interesting but I admit I have my doubts with all of them. Adding one to these speculations: could the intensive stretching cause changes in the optimal joint configuration for force development (length-force relation)? This may be related to what you mention on line 285

What I missed in the discussion: could the authors speculate about a potential leveling off of this stretch-induced strength increase?  Eight weeks lead to a 7% increase in strength, do you believe that if the program would continue for another 8 weeks, the strength increases would be larger? Any indications in the literature for this? This is important, since a 7% increase may be large in statistical terms, but he practical relevance seems limited: it only is 7 % (I know, for weak subjects, this may be important, but still 7 % isn’t a lot)

Please rephrase (or better remove) the following sentence from the practical implications: Enhancing strength capacity seems of high relevance for athletic performance 297 due to its relationship to several specific movement tasks such as jumping and sprinting 298 [46] or throwing ball velocity in handball [47]. I think it is not prudent to couple the present results found in untrained subjects with athletic performance in general.

Line 317 is very important: Nevertheless, this study used just one given elbow angle, 317 which may be of limited validity for other joint angles.

It is very challenging to standardize bench press execution, especially in relatively untrained subjects

Other:

Table 1 p² = 0,448  Typo error 0.448

Table 2: The units for force development seem missing (Ns, I assume)

Author Response

Dear Reviewer 2,

Thank you for your quick response. All changes are highlighted in yellow.
In the following, we reply to you comment in detail.

Please indicate what the adherence was in supervised and unsupervised conditions. This is important since adherence always is a problem.

  • Thank you for your comment. In the supervised stretching group, each stretching session was conducted as an individual session with an examiner. For this purpose, a total of 30 minutes, including setup and teardown, was allocated for each 15-minute stretching session. The home stretching group received a band after being instructed on the exercises and performed the exercises at home. The training sessions were recorded using a protocol.
  • We give some additional information about the adherence of the supervised and unsupervised group. Please see Line 146 and 154 – 157.

How were the participants instructed to determine (increase) stretch intensity over time (both within a session as well as across the 8 weeks). In the discussion, I read that such an increase in intensity did not occur in the at-home situation…..this seems remarkable, I missed the mentioning of a placebo effect. I think this should be mentioned, although the authors probably are convinced that the effect is ‘real’, placebo effects also are real and cannot be ruled out

  • Thank you for the comment. The HBS group was instructed to maintain a maximum stretching during the stretching training. Since the stretching exercises in the home-based group were switched after 5 minutes, the stretching intensity for the exercise was adjusted after 5 minutes to the maximum subjective pain sensation. We have added information that the stretching intensity in both groups was up to the maximum tolerable pain threshold. Please see line 156-157
  • It is possible that the stretching intensity of the home-based group was lower, as the intensity was controllable through self-practice of the exercises. However, a placebo effect cannot be ruled out. We have added information regarding this limitation. Please see line 357 - 360

I read the potential mechanisms with interest and attention: Indeed, interesting but I admit I have my doubts with all of them. Adding one to these speculations: could the intensive stretching cause changes in the optimal joint configuration for force development (length-force relation)? This may be related to what you mention on line 285

  • Thank you for your comment. Panidi et al (2023) demonstrated a change in muscle length, but not in the pennation angle. We have added the possible influence of a change in joint configuration into the script. Please see line 306 - 310

What I missed in the discussion: could the authors speculate about a potential leveling off of this stretch-induced strength increase?  Eight weeks lead to a 7% increase in strength, do you believe that if the program would continue for another 8 weeks, the strength increases would be larger? Any indications in the literature for this? This is important, since a 7% increase may be large in statistical terms, but he practical relevance seems limited: it only is 7 % (I know, for weak subjects, this may be important, but still 7 % isn’t a lot)

  • Thank you. The success of stretching training depends on the total stretching time, the duration of stretching within a single stretching training, and the intensity of stretching. As far as we know, there is no literature on this. We added an information. Please see line 338 - 343

Please rephrase (or better remove) the following sentence from the practical implications: Enhancing strength capacity seems of high relevance for athletic performance 297 due to its relationship to several specific movement tasks such as jumping and sprinting 298 [46] or throwing ball velocity in handball [47]. I think it is not prudent to couple the present results found in untrained subjects with athletic performance in general.

  • Thank you for the comment. We rephrase this sentence. Please see line 321 – 326.

Line 317 is very important: Nevertheless, this study used just one given elbow angle, 317 which may be of limited validity for other joint angles.

  • Thank you for your comment. We added an information about the limited validity of other joint angles. Please see line 351 – 355.

It is very challenging to standardize bench press execution, especially in relatively untrained subjects

  • Thank you. During the familiarization phase, particular attention was paid to standardize the bench press movement execution. We added an information about the bench press execution. Please see line 78 – 81.

Table 1 ⴄp² = 0,448  Typo error 0.448

  • Thank you. We corrected it. Please see Table 2.

Table 2: The units for force development seem missing (Ns, I assume)

  • Thank you. Yes, it should be N*s. We corrected that. Please see table 3.

Best Regards

The Authors                    

Reviewer 3 Report

Comments and Suggestions for Authors

Comparison of supervised versus self- administered stretching on bench press maximal strength and force development

 Very interesting, actual and beneficial work. Everything related to human health and sport performance is important and socially significant. The manuscript structure in my opinion is excellent and it is obvious that the Authors have done a lot of work. Personally, I learned important information.

 1. Introduction

This paragraph has everything necessary to acquaint the reader with the background of the study and defines its tasks. In my opinion the aim of study can be described more clearly.

2. Materials and Methods

The sample of people included in the study is large enough. This indicates that the manuscript suggests depth and validity of the findings. It will be better and more clear the participant’s characteristics (height, weight etc.) to be presented in a table.

 3. Results

In my opinion this section is excellent presented in accordance to the obtained data structure. This work has its limitations which are described here.

 4. Conclusions

Good work and accepted.

 5. The Abstract and Conclusions can be improved.

Improvement. In the Abstract and in the Conclusions must be underline clearly the new results obtained from the authors which differ from those obtained till now. It must be underline the main authors contributions.

I hope that the proposed corrections will increase the quality of the manuscript.

Author Response

Dear Reviewer 3,

Thank you for your quick response. All changes are highlighted in yellow.
In the following, we reply to you comment in detail.

Comparison of supervised versus self- administered stretching on bench press maximal strength and force development

 Very interesting, actual and beneficial work. Everything related to human health and sport performance is important and socially significant. The manuscript structure in my opinion is excellent and it is obvious that the Authors have done a lot of work. Personally, I learned important information.

  1. Introduction

This paragraph has everything necessary to acquaint the reader with the background of the study and defines its tasks. In my opinion the aim of study can be described more clearly.

  • Thank you for your comment. We added some information about the aim of the study. Please see line 64 – 65.

  1. Materials and Methods

The sample of people included in the study is large enough. This indicates that the manuscript suggests depth and validity of the findings. It will be better and more clear the participant’s characteristics (height, weight etc.) to be presented in a table.

  • Thank you. We added a table of the participants characteristics. Please see line 99.

  1. Results

In my opinion this section is excellent presented in accordance to the obtained data structure. This work has its limitations which are described here.

  • Thank you for your comment!

  1. Conclusions

Good work and accepted.

  • Thank you again for your comment!

  1. The Abstract and Conclusions can be improved.

Improvement. In the Abstract and in the Conclusions must be underline clearly the new results obtained from the authors which differ from those obtained till now. It must be underline the main authors contributions.

I hope that the proposed corrections will increase the quality of the manuscript.

  • Thank you for your comment. In the abstract, we are unfortunately limited in the word counts. In the conclusions we added some information and stated out the opening fore further practical applications. Please see line 364 – 374.

Best regards

The Authors